# Explainable and Efficient Randomized Voting Rules

**Soroush Ebadian**
University of Toronto
soroush@cs.toronto.edu

**Aris Filos-Ratsikas**
University of Edinburgh
Aris.Filos-Ratsikas@ed.ac.uk

**Mohamad Latifian**
University of Toronto
latifian@cs.toronto.edu

**Nisarg Shah**
University of Toronto
nisarg@cs.toronto.edu

## Abstract

With a rapid growth in the deployment of AI tools for making critical decisions (or aiding humans in doing so), there is a growing demand to be able to explain to the stakeholders how these tools arrive at a decision. Consequently, voting is frequently used to make such decisions due to its inherent explainability. Recent work suggests that using randomized (as opposed to deterministic) voting rules can lead to significant efficiency gains measured via the distortion framework. However, rules that use intricate randomization can often become too complex to explain to the stakeholders; losing explainability can eliminate the key advantage of voting over black-box AI tools, which may outweigh the efficiency gains.

We study the efficiency gains which can be unlocked by using voting rules that add a simple randomization step to a deterministic rule, thereby retaining explainability. We focus on two such families of rules, randomized positional scoring rules and random committee member rules, and show, theoretically and empirically, that they indeed achieve explainability and efficiency simultaneously to some extent.

## 1 Introduction

In the past decade, AI and machine learning solutions have been deployed ubiquitously to make increasingly critical decisions that affect human lives. Consequently, there is a growing demand for these models and their decisions to be *explainable* [1, 2]. The literature makes a distinction between two types of explanations: *outcome explanations*, which explain to the stakeholders why the chosen outcome was selected in a given instance, and *procedural explanations*, which explain to the stakeholders the procedure of choosing outcomes across all possible instances.Much of the explainable AI (XAI) literature focuses on outcome explanations because many black-box AI solutions used in practice are too complex to admit simple procedural explanations [3].

However, there are several drawbacks of outcome explanations. First, it opens up the possibility of post-hoc explanations for why an outcome was selected. These are susceptible to adversarial reasoning that hides biases [4]. Also, psychological research suggests that people's perception of fairness of an outcome depends not only on the outcome itself, but also on the process by which the outcome is selected [5, 6], and the same outcome may be perceived as fair or unfair depending on the process used [6]. This motivates the need for *procedural explanations*. Note that an intuitive explanation of the procedure to select outcomes already serves as a rudimentary justification for why a given outcome was selected.

To that end, we turn our attention to voting. While explainability is a nascent demand in the AI ecosystem, voting rules, historically deployed for political decision-making, have always battled with the need to be able to explain to the voters how the winner of an election is chosen. Thus,

37th Conference on Neural Information Processing Systems (NeurIPS 2023).

most prominent voting rules admit intuitive procedural explanations. Due to this key advantage over black-box AI solutions, they have been used to automate decision-making in a variety of applications such as designing recommender systems [7, 8], information extraction [9], collaborative filtering [10], ensemble learning [11], and game-playing by AI agents [12]. The advantage is more apparent when the decisions at hand are more significant. For example, Noothigattu et al. [13] and Kahng et al. [14] propose the design of a voting-based virtual democracy system that can automate ethical decision-making in AI systems. When Lee et al. [15] applied this framework to automate the distribution of food donations, they found that their stakeholders appreciated the fact that voting-based decision-making *"embodied democratic values"* and being able to provide easy explanations *"allowed them to understand how algorithmic recommendations were made"*.

However, most prominent voting rules are deterministic because their primary use case is making infrequent, high-stakes democratic decisions, for which randomization is generally unpalatable [16]. Aside from selected applications such as forming citizens' assemblies, juries, and independent redistricting commissions, lottery is seldom used to select representatives [17]. But in AI applications, it is common to make frequent, low-stakes decisions, for which randomization is well-suited.

Research on voting theory suggests that allowing the voting rule to randomize has numerous benefits. It can be essential for avoiding the tyranny of the majority and guarantee minority representation [18, 19]. Randomization also acts as a barrier to manipulations by strategic agents by circumventing the Gibbard-Satterthwaite impossibility [20, 21]. Most importantly, it can unlock significant efficiency gains over using deterministic voting rules [22, 23, 19]. Unfortunately, randomized voting rules designed to optimize for efficiency can be highly complex and rely on intricate mathematical results such as the minimax theorem [19], making them difficult to explain to the end users.

In view of this, we explore the use of *explainable randomized voting rules* for improving the efficiency of automated decision-making. To ensure explainability, one possibility is to use only extremely simple randomized rules such as random dictatorship, where the most preferred option of a randomly chosen agent is selected. But this may leave significant possible efficiency gains on the table. Instead, we propose a hybrid approach that adds a simple (and thus, explainable) randomization step to well-understood deterministic decision-making processes. This drives our main research question:

> *What efficiency gains can be unlocked by explainable randomized voting rules which add a simple randomization step to deterministic voting rules?*

As a yardstick for efficiency, we turn to the *distortion framework* [24]. Proposed by Procaccia and Rosenschein [25], this framework posits that votes submitted by agents, typically rankings over a set of alternatives, are induced by more expressive preferences underneath, typically cardinal utility functions over the alternatives. In this framework, the goal of a randomized voting rule is to choose a lottery over the alternatives that minimizes *distortion*, the worst-case ratio between maximum social welfare (total utility to the agents) and that of the chosen lottery.

We study the distortion of two families of randomized voting rules, which we refer to as *randomized positional scoring rules* and *random committee member rules*. The former builds on the widely-popular family of (deterministic) positional scoring rules, under which each agent assigns a score to each alternative based on its position in her ranking. But instead of deterministically selecting an alternative with the highest total score, each alternative is selected with probability proportional to its score. In contrast to picking alternatives with varying probabilities, the latter family utilizes the simplicity of uniform randomization by picking, uniformly at random, a member of a subset of alternatives chosen deterministically. Our inspiration for these two families stems from the use of such rules by Boutilier et al. [22] and Ebadian et al. [19].

Let us illustrate an example rule from each family based on the popular Borda count method, in which scores of $m-1, \ldots, 0$ are assigned to ranks $1, \ldots, m$ respectively. A rule from the first family would pick each alternative with probability proportional to its total Borda score, while a rule from the second family may select uniformly at random among the $k$ alternatives with the highest Borda scores, for some fixed $k$. We select these two families of randomized voting rules because they admit straightforward procedural explanations. For instance, the aforementioned rules based on Borda count can be explained as follows (using version (a) for the former and (b) for the latter using $k = 3$).

> *"Each user gives zero points to their least preferred option, one point to the next best option, two points to the next best option, and so on. Points are tallied and...*

Table 1: Distortion and minimum welfare of common randomized positional scoring rules.

| | Plurality | Borda | Harmonic | Veto | $k$-Approvals $k \in [1, m^{1/3}]$ | $k \in [m^{1/3}, \sqrt{m}]$ | $k \in [\sqrt{m}, m]$ |
|---|---|---|---|---|---|---|---|
| dist | $\Theta(m\sqrt{m})$ | $\Theta(m^{5/4})$ | $\Theta(\sqrt{m}H_m)$ | $\Theta(m)$ | $\Theta\big(\frac{m\sqrt{m}}{k\sqrt{k}}\big)$ | $\Theta(m)$ | $\Theta(k\sqrt{m-k+1})$ |
| min-sw | $\Theta\big(\frac{n}{m^2}\big)$ | $\Theta\big(\frac{n}{m\sqrt{m}}\big)$ | $\Theta\big(\frac{n}{mH_m}\big)$ | $\Theta\big(\frac{n}{m}\big)$ | $\Theta\big(\frac{nk}{m^2}\big)$ | | $\Theta\big(\frac{n}{k(m-k+1)}\big)$ |

*(a) the chances of each option being selected are proportional to its total points.*
*(b) the three options with the highest total points are selected with an equal chance."*

Unlike prior work on distortion, which is often focused on identifying the *most efficient* rule, we provide a refined analysis that characterizes the distortion of many interesting rules in these families, allowing the system designer to pick the one most suited to the application at hand.

## 1.1 Our Results

**Randomized positional scoring rules.** We develop a whole swathe of novel techniques for analyzing distortion, and use them to obtain tight distortion (dist) bounds for randomized versions of well-known positional scoring rules such as plurality, Borda count, harmonic, veto, and $k$-approval, presented in Table 1. For comparison, we remark that the distortion of the best possible deterministic voting rule is $\Theta(m^2)$ [26, 23] and the best possible randomized voting rule is $\Theta(\sqrt{m})$ [19, 22]. For randomized positional scoring rules, the best bound is $\Theta(\sqrt{m \log m})$ due to [22]. En route to the distortion bounds shown in Table 1, we also obtain tight bounds for another useful efficiency metric, *minimum welfare* (min-sw), defined in Section 3.1. In the supplementary material, we apply our novel techniques to analyze the distortion of *randomized multi-level approval rules*, which are uniform mixtures of different randomized $k$-approval rules. We demonstrate the strength of this result by using it to derive tight distortion bounds for a recently studied randomized positional scoring rule due to Gkatzelis et al. [27].

Our distortion bound for the randomized plurality rule (better known as random dictatorship) may be of independent interest because it is a widely studied voting rule [28–30]. It is also fascinating that the distortion of randomized $k$-approval is highly non-monotone: first decreasing from $\Theta(m\sqrt{m})$ to $\Theta(m)$ when $k$ grows from 1 to $m^{1/3}$, then staying $\Theta(m)$ when $k$ grows further to $\sqrt{m}$, then increasing again to $\Theta(m\sqrt{m})$ by $k = m - \Theta(m)$, and finally decreasing again to $\Theta(m)$ by $k = m$.

**Random committee member rules.** For the family of random committee member rules, we design a novel voting rule, which selects a random member of a *top-biased stable $k$-committee*, and achieves a distortion of $O(\max\{k, m^2/(k\sqrt{k})\})$. We complement this with a lower bound of $\Omega(\max\{k, m^2/k^2\})$ on the distortion of any rule in this class.

**Experiments.** Our experiments with synthetic data indicate that while the (worst-case) distortion of various rules in both families is not significantly better than the optimal deterministic distortion of $\Theta(m^2)$ and often even worse than the $\Theta(m)$ distortion of the trivial rule selecting a uniformly random alternative, rules from both families (almost always) significantly outperform deterministic voting rules and randomized positional scoring rules (almost always) outperform the uniform-random rule as well. This suggests that one should strongly consider replacing deterministic rules with explainable randomized rules from these two families in order to achieve significantly improved efficiency.

## 1.2 Additional Related Work

**Outcome explanations in voting.** We focus on (randomized) voting rules with procedural explanations because that is, in our view, the key advantage of voting over black-box AI solutions. Nonetheless, there is also compelling literature on producing outcome explanations for voting rules. Classical work that seeks voting rules satisfying qualitative axioms such as Condorcet consistency can be viewed in this light. However, these axioms often provide a justification only in limited instances with a special structure. Cailloux and Endriss [31] propose a method for producing a justification on any given instance by starting from an axiomatic justification on a nearby special instance and using a chain of explanations relating adjacent pairs of instances to arrive at the given instance. Peters et al.

[32] bound the length of such chains, focusing especially on positional scoring rules, while Boixel and Endriss [33] and Boixel et al. [34] study computational aspects of finding them.

**On randomized positional scoring rules.** The family of randomized positional scoring rules was first proposed by Barbera [35], under the name of point-voting schemes. He establishes several appealing properties of these rules, including strategyproofness (i.e., no agent can ever strictly benefit from misreporting her preferences). Note that the outcome of a randomized positional scoring rule can be computed by first selecting an agent uniformly at random, and then, for each $k$, selecting her $k$-th most preferred alternative with probability proportional to score assigned to position $k$. In this sense, implementing a randomized positional scoring rule requires little elicitation.

**On distortion.** We use the utilitarian distortion framework proposed by Procaccia and Rosenschein [25] and developed by Boutilier et al. [22], where agents have unit-sum utilities over alternatives. This framework has been applied to a broad range of settings such as committee selection [36], participatory budgeting [37], distributed elections [38], elicitation-efficiency tradeoff [39–43], and matching [44, 40]. Our methodological novelty is that we are the first to provide non-trivial absolute guarantees on the minimum welfare achieved across all instances, which is a reasonable efficiency measure in itself, and in turn use them to achieve tight distortion bounds (which compare the welfare achieved on each instance to the optimal welfare in that instance). We also note that there is a related framework of metric distortion [45], which assumes that agents have costs for the alternatives induced by an underlying metric space in which they are both embedded. This framework utilizes the restriction imposed by the triangle inequality instead of a unit-sum normalization. We refer the reader to the survey of Anshelevich et al. [24] for a detailed overview.

## 2 The Setting

Let us formally introduce our setting and define the notion of distortion. We will define each class of explainable voting rules in the section in which we will study it. We use the terminology of *elections* for consistency with the literature, but our setting captures a general decision-making scenario in which one of several alternatives must be chosen by aggregating conflicting preferences or opinions.

**Basic notation.** Let $[t] = \{1, 2, \ldots, t\}$ for $t \in \mathbb{N}$, and define $\Delta(S)$ to be the probability simplex over the finite set $S$. For a vector $\vec{s} = (s_1, \ldots, s_t)$, denote its $\ell^1$-norm by $\|\vec{s}\|_1 = \sum_{i \in [t]} s_i$.

**Utilitarian voting.** A (single-winner) election consists of sets $N = [n]$ of $n$ agents and $A = [m]$ of $m$ alternatives. Each agent $i \in N$ has a personal *cardinal* utility function $u_i \colon A \mapsto \mathbb{R}_{\geqslant 0}$, where $u_i(a)$ is the value associated by agent $i$ to alternative $a$. Following the convention in the literature (e.g., see [22, 19]), we adopt the *unit-sum* normalization of utility functions: for every $i \in N$, let $\sum_{a \in A} u_i(a) = 1$. Aziz [46] provides several compelling justifications for using unit-sum utility functions. For a utility profile $\vec{u} = (u_1, \ldots, u_n)$ and a subset of agents $T \subseteq N$, define the *social welfare* of an alternative $a \in A$ with respect to $T$ as $\mathsf{sw}_T(a, \vec{u}) = \sum_{i \in T} u_i(a)$. We write $\mathsf{sw}_N$ simply as $\mathsf{sw}$, and drop $\vec{u}$ when it is clear from the context. As an extension, for a distribution $p \in \Delta(A)$ over the alternatives, define $u_i(p) = \mathbb{E}_{a \sim p}[u_i(a)]$ and its social welfare as $\mathsf{sw}(p, \vec{u}) = \sum_{i \in N} u_i(p)$. Our goal is to find a distribution over alternatives with high social welfare. We will sometimes construct and analyze a *partial utility profile*, where the utilities of each agent sum to *at most* 1.

**Ordinal preferences and voting rules.** We consider *voting rules* that have access only to the *ordinal* preferences induced by the utilities. This is because a ranking of alternatives can often be elicited with less cognitive burden or estimated more accurately than exact numerical utilities for each alternative.

Each agent $i \in N$ submits a preference ranking $\sigma_i \colon [m] \mapsto A$ of the alternatives. We use $\mathsf{rank}_i(a) = \sigma_i^{-1}(a)$ to denote the rank of alternative $a$ in agent $i$'s preference ranking (the most preferred alternative has rank 1), and $a \succ_i a'$ to denote that agent $i$ prefers $a$ to $a'$ (i.e., $\mathsf{rank}_i(a) < \mathsf{rank}_i(a')$). We assume that $\sigma_i$ is consistent with agent $i$'s utility function $u_i$, i.e., $a \succ_i a'$ implies $u_i(a) \geqslant u_i(a')$ for all $a, a' \in A$; ties can be broken arbitrarily without affecting our distortion upper bounds.

Let $\vec{\sigma} = (\sigma_i)_{i \in N}$ be a *preference profile* and $\mathcal{C}(\vec{\sigma})$ denote the set of utility profiles $\vec{u}$ such that $\sigma_i$ is consistent with $u_i$ for each agent $i \in N$. A voting rule $f$ takes a preference profile $\vec{\sigma}$ as input and returns a distribution $p$ over alternatives.

**Distortion.** The distortion of a distribution $p \in \Delta(A)$ over alternatives with respect to a utility profile $\vec{u}$ is defined as

$$\mathsf{dist}(p, \vec{u}) = \frac{\max_{a \in A} \mathsf{sw}(a, \vec{u})}{\mathsf{sw}(p, \vec{u})}.$$

The distortion of a voting rule $f$ is defined as its worst-case distortion over all instances: $\mathsf{dist}_m(f) = \sup_{\vec{\sigma}, \vec{u} \in \mathcal{C}(\vec{\sigma})} \mathsf{dist}(f(\vec{\sigma}), \vec{u})$, where the supremum is taken over all instances with $m$ alternatives and any number of agents. For simplicity, we drop $m$ and write $\mathsf{dist}(f)$.

## 3 Distortion of Randomized Positional Scoring Rules

The first class of explainable randomized voting rules we study is randomized positional scoring rules, or point-voting schemes [35]. This builds on the popular class of (deterministic) positional scoring rules, which assign scores to alternatives based on their positions in agents' preference rankings, and adds an easy-to-explain randomization step where each alternative is chosen with probability proportional to its score instead of deterministically choosing the one with the highest score.

**Positional scoring rules.** A *scoring vector* $\vec{s} = (s_1, \ldots, s_m)$ assigns a score $s_r$ to each position $r \in [m]$ and satisfies $s_1 \geqslant s_2 \geqslant \ldots \geqslant s_m \geqslant 0$. For an alternative $a \in A$, let $\mathsf{score}_i(a, \vec{s}) = s_{\mathsf{rank}_i(a)}$ be the score $a$ obtains from agent $i$, and for $N' \subseteq N$, $\mathsf{score}_{N'}(a, \vec{s}) = \sum_{i \in N'} \mathsf{score}_i(a, \vec{s})$. Note that $\sum_{a \in A} \mathsf{score}_N(a, \vec{s}) = n \cdot \|\vec{s}\|_1$. We drop $\vec{s}$ when it is clear from the context. For a scoring vector $\vec{s}$, we can define the following rules.

- The *deterministic positional scoring rule* $f_{\vec{s}}^{\mathsf{det}}$ selects the top scored alternative (breaking ties arbitrarily), i.e., $f_{\vec{s}}^{\mathsf{det}}(\vec{\sigma}) = \arg\max_{a \in A} \mathsf{score}_N(a, \vec{s})$.
- The *randomized positional scoring rule* $f_{\vec{s}}^{\mathsf{rand}}$ selects every alternative $a \in A$ with probability proportional to its score, i.e., $\Pr[f_{\vec{s}}^{\mathsf{rand}}(\vec{\sigma}) = a] = \mathsf{score}_N(a, \vec{s}) / (n \cdot \|\vec{s}\|_1)$.

The deterministic rules introduced above include several well-known voting rules such as *plurality*, *Borda*, *k-approval*, *veto*, and *harmonic* defined, by the following scoring vectors, respectively:

$$\vec{s}_{\mathsf{plu}} = (1, 0, \ldots, 0), \qquad \vec{s}_{\mathsf{Borda}} = (m-1, m-2, \ldots, 0), \quad \vec{s}_{k\text{-approval}} = (\underbrace{1, \ldots, 1}_{k \text{ ones}}, 0, \ldots, 0),$$

$$\vec{s}_{\mathsf{veto}} = (1, 1, \ldots, 1, 0), \quad \vec{s}_{\mathsf{harmonic}} = (1, 1/2, 1/3, \ldots, 1/m).$$

We refer to the randomized versions of these rules as "randomized $f$", where $f \in \{\mathsf{plurality}, \mathsf{Borda}, \mathsf{harmonic}, \mathsf{veto}\}$, and extend this terminology to any positional scoring rule $f_{\vec{s}}$. Note that "randomized plurality" is more widely known as *random dictatorship* (see, e.g., [28, 29]).

### 3.1 High-Level Distortion Analysis and Novel Insights

**Logarithmic rounding of the scores.** Our first useful insight is that we can reduce the number of distinct scores by rounding any score down to the nearest power of $1 + \alpha$, for a constant $\alpha > 0$, and this only changes the distortion of the rule by a factor of at most $1 + \alpha$.

**Lemma 1** (Rounding Down Scores). *Let $\alpha \geqslant 0$, and $\vec{s}, \vec{s'}$ be scoring vectors such that $s'_j \leqslant s_j \leqslant (1 + \alpha)s'_j$ for all $j \in [m]$. Then, for every preference profile $\vec{\sigma}$ and consistent utility profile $\vec{u} \in \mathcal{C}(\vec{\sigma})$,*

$$\frac{1}{1 + \alpha} \cdot \mathsf{sw}(f_{\vec{s'}}^{\mathsf{rand}}(\vec{\sigma}), \vec{u}) \leqslant \mathsf{sw}(f_{\vec{s}}^{\mathsf{rand}}(\vec{\sigma}), \vec{u}) \leqslant (1 + \alpha) \cdot \mathsf{sw}(f_{\vec{s'}}^{\mathsf{rand}}(\vec{\sigma}), \vec{u}),$$

*and consequently, $\frac{1}{1+\alpha} \cdot \mathsf{dist}(f_{\vec{s'}}^{\mathsf{rand}}) \leqslant \mathsf{dist}(f_{\vec{s}}^{\mathsf{rand}}) \leqslant (1 + \alpha) \cdot \mathsf{dist}(f_{\vec{s'}}^{\mathsf{rand}})$.*

By applying this transformation, scores in the range $[\|\vec{s}\|_1/(4m^2), \|\vec{s}\|_1]$ can be reduced to $O(\log m)$ distinct values, which we will find helpful in the subsequent sections. In the supplementary material, we show that by ignoring the remaining scores by reducing any $s_j \leqslant \|\vec{s}\|_1/(4m^2)$ to 0 changes the distortion further by another factor of at most two. Hence, we can limit our focus to scoring vectors that contain $O(\log m)$ distinct scores, resulting in only a constant factor loss in the distortion analysis.

**High-level distortion analysis.** After reducing the scoring vector to $O(\log m)$ distinct scores, we partition the agents into $O(\log m)$ groups based on where they rank the optimal alternative $a^*$. This is only for the analysis; the voting rule does not know the optimal alternative. More formally, let

$0 = \ell_0 < \ell_1 < \ldots < \ell_q = m$ be the indices where the score changes in the reduced scoring vector, where, for each $r \in [q]$, we have $s_i = s_{\ell_r}$ for all $i \in [\ell_{r-1}+1, \ell_r]$, and $s_{\ell_1} > \ldots > s_{\ell_q}$. Furthermore, let $N_r$ be the set of agents who rank $a^*$ among positions $[\ell_{r-1} + 1, \ell_r]$.

Next, borrowing an insight from the prior distortion literature [37, 42, 43], we round the agent utilities to the nearest power of two and ignore utilities below $1/m^2$, reducing the number of distinct utility values to $O(\log m)$ while losing at most a constant factor in the analysis. This allows us to subdivide voters in each group $N_r$ into $O(\log m)$ subgroups such that every agent in a subgroup with the same utility, say $\tau$, for $a^*$. We then employ three strategies to bound the distortion within each subgroup. Finally, we show that the overall distortion can be upper bounded, up to logarithmic factors, by the worst of these $O(\log^2 m)$ distortion bounds across all subgroups of all the $N_r$ groups.

**Strategy 1 (Welfare above a$^*$).** Voters in a subgroup of $N_r$ who have utility $\tau$ for $a^*$ also have utility at least $\tau$ for their top $\ell_{r-1}$ alternatives. This helps us derive social welfare guarantees for the randomized positional scoring rule.

**Strategy 2 (Probability of a$^*$).** We use the well-known observation (see, e.g., [22]) that the distortion is always upper bounded by the inverse of the probability of selecting $a^*$.

**Strategy 3 (Absolute Welfare Lower Bounds).** Another novel insight from our work is that proving an absolute lower bound on the welfare achieved by a rule across all instances can be useful in bounding the distortion, even though the latter needs to compare the welfare achieved in each instance to the optimum welfare in that instance. We carefully analyze and approximate, up to logarithmic factors, the minimum welfare achieved by the randomized positional scoring rules we study, and use it to bound its distortion. A similar idea has been used in other domains (see, e.g. [47]), but to the best of our knowledge, we are the first to successfully apply it to distortion analysis.

**Definition 1** (Minimum Welfare). *Define the* minimum welfare *of a distribution over alternatives $p \in \Delta(A)$ on a preference profile $\vec{\sigma}$ as* $\mathsf{min\text{-}sw}(p, \vec{\sigma}) = \inf_{\vec{u} \in \mathcal{C}(\vec{\sigma})} \mathsf{sw}(p, \vec{u})$*, which is the minimum social welfare of $p$ across all consistent utility profiles. The* minimum welfare *of a voting rule $f$ is the minimum welfare of its output, minimized over all preference profiles:* $\mathsf{min\text{-}sw}_{n,m}(f) = \min_{\vec{\sigma}} \mathsf{min\text{-}sw}(f(\vec{\sigma}), \vec{\sigma})$*, where the minimum is taken over all preference profiles with $n$ agents and $m$ alternatives. We drop $n$ and $m$ when clear from the context.*

Due to $\mathcal{C}(\vec{\sigma})$ being compact, the infimum in the $\mathsf{min\text{-}sw}(p, \vec{\sigma})$ definition is indeed attained. In the supplementary material, we make an structural observation that for any preference profile and distribution $p \in \Delta(A)$, the minimum welfare is at most $n/m$, attained at a *dichotomous utility profile*. Furthermore, every randomized positional scoring rule $f_{\vec{s}}^{\mathsf{rand}}$ satisfies $\mathsf{min\text{-}sw}(f_{\vec{s}}^{\mathsf{rand}}) \in [n/(4m^2), n/m]$. We also show how to approximate minimum welfare better (up to constants or logarithmic terms); see Table 1 for tight bounds for the rules induced by common scoring vectors.

To this end, we present our most intricate technical lemma to derive a generic welfare lower bound, which we use to apply Strategies 1 and 3. Instead of focusing only on $\Pr[f_{\vec{s}}^{\mathsf{rand}}(\vec{\sigma}) = a^*]$, it bounds the overall welfare expression $\sum_{a \in A} \Pr[f_{\vec{s}}^{\mathsf{rand}}(\vec{\sigma}) = a] \cdot \mathsf{sw}_T(a)$; the product of probability of selection and welfare of an alternative leads to a quadratic program, where the variables encode the worst case, and this is analytically solved using the Karush-Kuhn-Tucker (KKT) conditions.

**Lemma 2.** *Fix any scoring vector $\vec{s}$, preference profile $\vec{\sigma}$, subset of agents $T \subseteq N$, threshold $\tau \geqslant 0$, and rank $\ell \in [m]$. For a partial utility profile $\vec{u}$ in which every agent in $T$ has utility at least $\tau$ for each of her top $\ell$ alternatives and all other utilities are $0$, we have:*

$$\mathsf{sw}_T(f_{\vec{s}}^{\mathsf{rand}}(\vec{\sigma}), \vec{u}) \geqslant \tau \cdot \frac{|T|\ell}{2n\|\vec{s}\|_1} \min_{h \in [m]} \frac{1}{h}\left(2s_\ell \cdot |T|\ell + (n - |T|) \cdot \sum_{j=1}^{h} s_{m-j+1}\right).$$

Instead of tediously explaining the lemma, we will later show how its straightforward application wondrously gives us the desired welfare lower bound for the example of randomized Borda rule.

## 3.2 Analyzing Common Rules

We are ready to present our main result, which uses the aforementioned insights to pinpoint the asymptotic distortion of common randomized positional scoring rules.

**Theorem 2.** *For $f \in \{$plurality, Borda, harmonic, veto, $k$-approvals$\}$, the minimum welfare* ($\mathsf{min\text{-}sw}$) *and the distortion* ($\mathsf{dist}$) *of the 'randomized $f$' rule are as shown in Table 1.*

Due to space limitations, we only provide a proof for the distortion upper bound of the randomized Borda rule, and defer the rest to the supplementary material. For conciseness, we use the notation $\mathsf{Borda}(a) \triangleq \mathsf{score}(a, \vec{s}_{\mathsf{Borda}})$. First, we need the following lower bound on its minimum welfare.

**Lemma 3.** *The minimum welfare of the randomized Borda rule is* $\mathsf{min}\text{-}\mathsf{sw}(f_{\vec{s}_{\mathsf{Borda}}}^{\mathsf{rand}}) = \Omega(\frac{n}{m\sqrt{m}})$.

*Proof.* Fix any preference profile $\vec{\sigma}$ and consistent utility profile $\vec{u} \in \mathcal{C}(\vec{\sigma})$. Our goal is to show that $\mathsf{sw}(f_{\vec{s}_{\mathsf{Borda}}}^{\mathsf{rand}}(\vec{\sigma}), \vec{u}) = \Omega(\frac{n}{m\sqrt{m}})$. First, we make a few modifications to the scoring vector and the preference profile that are guaranteed to not increase the welfare, and then invoke Lemma 2.

*Simplify the scores.* Let us consider the scoring vector $\vec{s}'$ which is equal to $\vec{s}_{\mathsf{Borda}}$ except the top $m/2$ scores are all equal to $m/2$. Note that $s'_j \leqslant (s_{\mathsf{Borda}})_j \leqslant 2s'_j$ for all $j \in [m]$. Hence, invoking Lemma 1 with $\alpha = 1$ yields $\mathsf{sw}(f_{\vec{s}_{\mathsf{Borda}}}^{\mathsf{rand}}(\vec{\sigma}), \vec{u}) \geqslant 1/2 \cdot \mathsf{sw}(f_{\vec{s}'}^{\mathsf{rand}}(\vec{\sigma}), \vec{u})$.

*Simplify the preference and utility profiles.* Next, let us lower bound $\mathsf{sw}(f_{\vec{s}'}^{\mathsf{rand}}(\vec{\sigma}), \vec{u})$. For each agent $i$, let $A_i$ be the set of top $m/2$ alternatives in $\sigma_i$ and $a_i \in \arg\min_{a \in A_i} \mathsf{score}(a, \vec{s}')$ be the alternative in $A_i$ with the lowest score (equivalently, probability of selection under $f_{\vec{s}'}^{\mathsf{rand}}(\vec{\sigma})$). Now,

$$u_i(f_{\vec{s}'}^{\mathsf{rand}}(\vec{\sigma})) \geqslant \sum_{a \in A_i} \frac{\mathsf{score}(a, \vec{s}')}{n\|\vec{s}'\|_1} \cdot u_i(a) \overset{(1)}{\geqslant} \frac{\mathsf{score}(a_i, \vec{s}')}{n\|\vec{s}'\|_1} \cdot \left(\sum_{a \in A_i} u_i(a)\right) \overset{(2)}{\geqslant} \frac{\mathsf{score}(a_i, \vec{s}')}{n\|\vec{s}'\|_1} \cdot \frac{1}{2},$$

where (1) follows from the definition of $a_i$ and (2) uses the fact that each agent has a total utility of at least $1/2$ for her top $m/2$ alternatives (due to the pigeonhole principle).

*Invoking Lemma 2.* The final expression above can be written as $u'_i(f_{\vec{s}'}^{\mathsf{rand}}(\vec{\sigma}'))$, where $\vec{\sigma}'$ is a preference profile in which each agent $i$ ranks $a_i$ first, and $\vec{u}'$ is a partial utility profile in which each agent $i$ has utility $1/2$ for her top alternative and $0$ for the rest. Summing the above for all agents, we have $\mathsf{sw}(f_{\vec{s}'}^{\mathsf{rand}}(\vec{\sigma}), \vec{u}) \geqslant \mathsf{sw}(f_{\vec{s}'}^{\mathsf{rand}}(\vec{\sigma}'), \vec{u}'))$. Thus, to lower bound it, we invoke Lemma 2 with $\vec{s} \leftarrow \vec{s}'$, $\vec{\sigma} \leftarrow \vec{\sigma}'$, $T$ being an arbitrary subset of $n/2$ agents, $\tau \leftarrow 1/2$, and $\ell \leftarrow 1$. Using $\|\vec{s}'\|_1 = \frac{m}{2} \cdot \frac{m}{2} + \binom{m/2}{2} = \frac{m(3m-2)}{8}$, this gives us

$$\mathsf{sw}(f_{\vec{s}'}^{\mathsf{rand}}(\vec{\sigma}'), \vec{u}') \overset{(1)}{\geqslant} \frac{1}{2} \cdot \frac{\frac{n}{2} \cdot 1}{2n \cdot \frac{m(3m-2)}{8}} \cdot \min_{h \in [m/2]} \frac{1}{h}\left(2 \cdot \frac{m}{2} \cdot \frac{n}{2} + \frac{n}{2} \cdot \frac{h(h-1)}{2}\right)$$

$$= \frac{n}{4m(3m-2)} \cdot \min_{h \in [m/2]} \left(\frac{2m}{h} + h - 1\right) \overset{(2)}{\geqslant} \frac{n \cdot (2\sqrt{2m} - 1)}{4m(3m-2)} \overset{(3)}{\geqslant} \frac{n}{6m\sqrt{m}},$$

where the restriction to $h \in [m/2]$ in (1) is based on the fact that the bound would be $\Omega(n/m)$ when $h > m/2$, (2) is due to the AM-GM inequality, and (3) uses $m \geqslant 2$. Connecting the dots, we have

$$\mathsf{sw}(f_{\vec{s}_{\mathsf{Borda}}}^{\mathsf{rand}}(\vec{\sigma}), \vec{u}) \geqslant \frac{1}{2} \cdot \mathsf{sw}(f_{\vec{s}'}^{\mathsf{rand}}(\vec{\sigma}), \vec{u}) \geqslant \frac{1}{2} \cdot \mathsf{sw}(f_{\vec{s}'}^{\mathsf{rand}}(\vec{\sigma}'), \vec{u}') \geqslant \frac{n}{12m\sqrt{m}}. \qquad \square$$

To translate the welfare lower bound from Lemma 3 into a distortion upper bound, we need the following relation between the Borda score of an alternative and its social welfare.

**Lemma 4.** *For any preference profile $\vec{\sigma}$, consistent utility profile $\vec{u} \in \mathcal{C}(\vec{\sigma})$, and alternative $a \in A$, we have* $\mathsf{sw}(a) \leqslant (\mathsf{Borda}(a) + n)/m$.

The desired distortion upper bound can now be derived using a standard analysis. The crux of our proof for the randomized Borda rule lies in our intricate derivation of its minimum welfare in Lemma 3, for which Lemma 2 does the heavy lifting.

**Lemma 5.** *The distortion of the randomized Borda rule is* $O(m^{5/4})$.

*Proof.* Fix any preference profile $\vec{\sigma}$ and consistent utility profile $\vec{u} \in \mathcal{C}(\vec{\sigma})$. Let $a^* \in \arg\max_{a \in A} \mathsf{sw}(a, \vec{u})$ be an optimal alternative. By Strategy 2 from Section 3.1, we know that the distortion is at most $\frac{n\|\vec{s}_{\mathsf{Borda}}\|_1}{\mathsf{Borda}(a^*)}$. Following Strategy 3 from Section 3.1 and the minimum welfare analysis in Lemma 3, we have that distortion is at most $\frac{\mathsf{sw}(a^*)}{n/(12m\sqrt{m})}$, which, using Lemma 4, is at

most $12\sqrt{m} \cdot \left( \mathsf{Borda}(a^*)/n + 1 \right)$. Putting everything together, and using the fact that $\min\{a, b\} \leqslant \sqrt{ab}$ for all $a, b \in \mathbb{R}_{\geqslant 0}$, the distortion is upper bounded by

$$\min \left\{ \frac{n\|\vec{s}_{\mathsf{Borda}}\|_1}{\mathsf{Borda}(a^*)}, \frac{\mathsf{Borda}(a^*) \cdot 12\sqrt{m}}{n} + 12\sqrt{m} \right\}$$

$$\leqslant 12\sqrt{m} + \sqrt{\frac{n \cdot m(m-1)/2}{\mathsf{Borda}(a^*)} \cdot \frac{\mathsf{Borda}(a^*) \cdot 12\sqrt{m}}{n}} \leqslant 8\sqrt{m} + \sqrt{6}m^{5/4} = O(m^{5/4}). \quad \square$$

## 4 Random Committee Member Rules

Next, we focus on our second class of explainable randomized voting rules that select an alternative uniformly at random from a shortlisted committee of size $k \in [m]$. We call them *random k-committee member* rules. For $k = 1$, we are left with deterministic rules, among which plurality achieves the optimal distortion of $\Theta(m^2)$. For $k = m$, we are left with uniform selection among all alternatives, which has distortion $\Theta(m)$. Is it possible that, for some intermediate value of $k$, we in fact achieve sublinear distortion? Could we achieve distortion at most logarithmic factors worse than the optimal $\Theta(\sqrt{m})$, like with randomized positional scoring rules? We answer the former positively but the latter negatively. First, we present a lower bound proving that any random $k$-committee member rule, for any value of $k$, incurs a distortion of at least $\Omega(m^{2/3})$.

**Theorem 3.** *For $k \in [m]$, any random k-committee member rule incurs $\Omega(\max(k, m^2/k^2))$ distortion. This lower bound is at least $\Omega(m^{2/3})$ for all k.*

As a result, this class of rules is less powerful than the class of randomized positional scoring rules, and thus, the class of all randomized rules. However, especially for small values of $k$, we gain the benefit of randomizing over a small support, which could translate to greater explainability.

To derive upper bounds, one might be tempted to turn again to positional scoring rules, and consider selecting uniformly at random from the $k$ alternatives with the highest score according to some scoring vector. In the supplementary material, we show that using plurality scoring vector yields $\Theta(m(m - k + 1))$ distortion. While it nicely interpolates between the extremes of $\Theta(m^2)$ at $k = 1$ and $\Theta(m)$ at $k = m$, it fails to achieve sublinear distortion, which we prove to be achievable. What about other scoring vectors? Unfortunately, it is relatively easy to see that using scoring vectors such as Borda, harmonic, or veto results in unbounded distortion. Despite the disappointing worst-case performance, we show in Section 5 that these rules perform relatively well empirically.

Next, we design a novel random $k$-committee member rule, which, with the right value of $k$, allows us to achieve sublinear distortion.

**Theorem 4.** *There is a polynomial-time computable random k-committee member rule with distortion $O(\max\{k, m^2/(k\sqrt{k})\})$. This is minimized at $k = m^{4/5}$, where the bound becomes $O(m^{4/5})$.*

To achieve the above, we concoct a three-way mixture of an *approximately stable committee* [48], a powerful notion which has been used to derive optimal randomized rules [19], alternatives with high plurality scores, and alternatives picked carefully to guarantee high minimum welfare from sufficiently many agents. We refer to the committee thus formed as a *top-biased stable k-committee*. The rule that returns this committee is presented as Algorithm 1 in the supplementary material.

Theorems 3 and 4 leave open the question of the optimal distortion that can be achieved by a random committee member rule, sandwiching this value between $O(m^{4/5})$ and $\Omega(m^{2/3})$. It is also interesting to wonder which value of $k$ is optimal. Our upper bound is optimized at $k = m^{4/5}$, and our lower bound implies that the optimal $k$ must be in $[m^{3/5}, m^{4/5}]$ as the distortion outside of that range is $\Omega(m^{4/5})$. See Section 5 for an empirical evaluation of the optimal $k$.

## 5 Experiments

Next, we empirically evaluate the efficiency of explainable rules studied in the previous sections.

**Rules.** We consider three classes of rules: deterministic positional scoring rules, randomized positional scoring rules from Section 3, and, from Section 4, rules that select uniformly at random from the $k$ alternatives with the highest scores (henceforth, *uniform random k-positional scoring rules*).

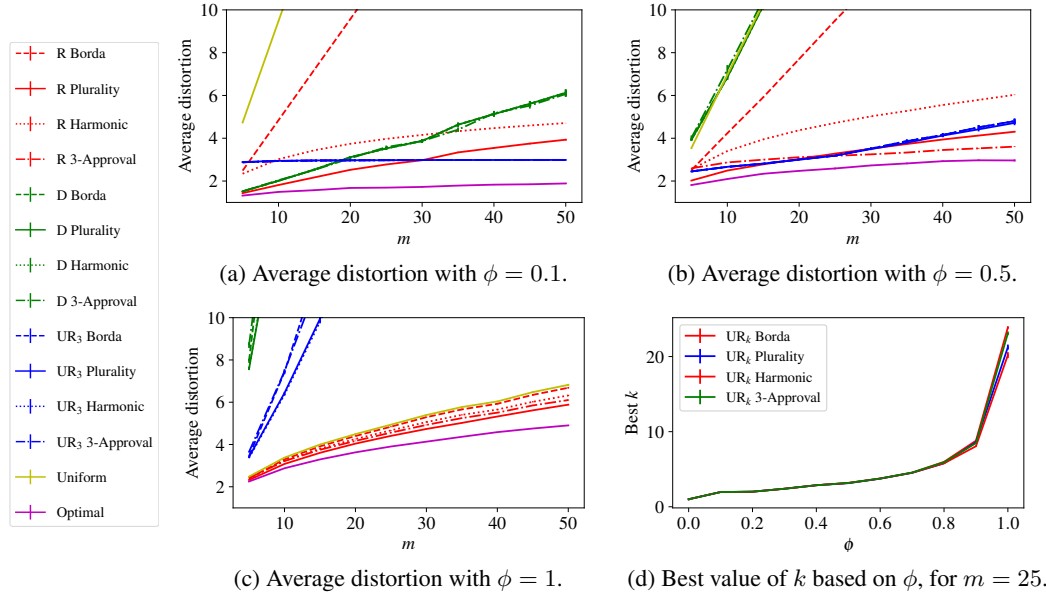

(a) Average distortion with $\phi = 0.1$.

(b) Average distortion with $\phi = 0.5$.

(c) Average distortion with $\phi = 1$.

(d) Best value of $k$ based on $\phi$, for $m = 25$.

Figure 1: All figures show results averaged over $150$ runs along with the standard error. Figures 1a to 1c share the legend on the left.

We consider four representative scoring vectors $f \in \{\text{Plurality}, \text{Borda}, \text{Harmonic}, \text{3-Approval}\}$, and denote the corresponding rules in the three classes by 'D $f$', 'R $f$', and 'UR$_k f$', respectively. Thus, overall, we test 12 voting rules. As benchmarks, we also add the *Uniform* rule, which selects an alternative uniformly at random from the set of all alternatives, and the *Instance Optimal* rule, which selects the lottery over alternatives minimizing distortion on the preference profile. Boutilier et al. [22] show how to use linear programming to compute the latter in polynomial time.

**Data Generation.** We generate preference profiles by sampling $n$ rankings over $m$ alternatives iid from the Mallows model [49], which is widely used in machine learning and statistics. The model takes as input an underlying reference ranking $\sigma^*$ (which can be set arbitrarily) and a *dispersion parameter* $\phi \in [0, 1]$. When $\phi = 1$, the model converges to a uniform distribution over all $m!$ rankings (also known as *impartial culture*), whereas $\phi \to 0$ converges to the point distribution where $\sigma^*$ is sampled with probability 1, so all the agents have the same preference ranking in the sampled profile. For a precise definition of the model and an efficient algorithm to sample from it (which we use in our experiments), see the work of Lu and Boutilier [50]. For each combination of $n = 100$ agents, $m \in \{5, 10, \ldots, 50\}$ alternatives, and dispersion parameter $\phi \in \{0, 0.1, \ldots, 1\}$, we sample 150 instances, and report averages along with the standard error. In the supplementary material, we report the results for two other statistical models namely the Polya-Eggenberger urn model [51] and the Plackett-Luce model [52, 53].

**Evaluation.** For each rule $f$ under consideration and each instance $\vec{\sigma}$, we evaluate the efficiency of the output $f(\vec{\sigma})$ by measuring its instance-specific distortion $\text{dist}(f(\vec{\sigma}), \vec{\sigma})$. Note that this still takes a worst case over the utility profiles consistent with $\vec{\sigma}$, but unlike in Sections 3 and 4 where we also take a worst case over $\vec{\sigma}$, here we compute the expected distortion over $\vec{\sigma}$ drawn from the Mallows model. Again, we compute $\text{dist}(f(\vec{\sigma}), \vec{\sigma})$ using the LP-based approach of Boutilier et al. [22].

**Results.** Figures 1a to 1c show the average distortion of different rules for $\phi \in \{0.1, 0.5, 1\}$, respectively, fixing $m = 25$. For large $\phi$ (impartial culture), the alternatives are spread out in agents' preference rankings. Thus, it becomes likely that some alternative is ranked slightly lower than another alternative by many agents, but has much higher social welfare in total. We see that in this case, randomized positional scoring rules outperform deterministic and random committee member rules as well as the uniform benchmark since it is more efficient to give a chance of winning to each alternative. As $\phi$ decreases to $0.5$ and agent rankings become somewhat correlated, random committee member rules start to outperform some of the randomized positional scoring rules (though randomized plurality and randomized 3-approval still perform quite well). But crucially, both families

of rules still outperform deterministic rules and the improved performance of random committee member rules now allows them to outperform the uniform benchmark as well. At $\phi = 0.1$, when agent preferences are highly correlated, deterministic rules gain some traction. Nonetheless, at least one rule from one of the two randomized classes still outperforms all deterministic rules (randomized plurality for low $m$ and any random committee member rule for high $m$). Overall, randomized plurality and randomized 3-approval perform reasonably well, and randomized plurality outperforms all four deterministic rules across almost all values of $m$ and $\phi$. The evidence suggests that one can almost always choose an explainable randomized rule that achieves better efficiency than deterministic rules, though the choice of the rule may have to depend on the setting at hand. A detailed comparison of rules in each class can be found in the supplementary material.

In the above experiments, for random committee member rules (specifically, the uniform random $k$-positional scoring rules), we use a committee size of $k = 3$. Figure 1d shows the best value of $k$ (one that yields the minimum distortion) for different scoring vectors as a function of $\phi$. It turns out that the best $k$ is indeed very small ($\leqslant 5$) unless $\phi$ is really close to $1$. Thus, $k = 3$ is a reasonable choice that helps our random committee member rules achieve high efficiency. Still, it is possible to further optimize the efficiency of these rules by pairing them with their corresponding optimal value of $k$; the resulting average distortion is presented in the supplementary material.

## 6 Limitations and Future Work

**Explainability.** We focus on the families of randomized positional scoring rules and random committee member rules as two examples of explainable randomized voting rules. While we argue in the introduction that these families admit intuitive procedural explanations and provide example explanations, checking whether stakeholders find these explanations reasonable and satisfactory in the context of a real-life application requires an in-depth investigation, possibly via user studies. Our work also treats explainability as a qualitative attribute, but different rules — even within the same family — may differ in the *degree* to which they are explainable. Quantifying the degree of explainability, both theoretically and empirically, remains to be tackled.

**Efficiency.** Our work uses distortion as a yardstick for efficiency, leaving open exciting technical questions. While our analysis of randomized multi-level approval rules in the supplementary material takes a step towards characterizing the distortion of all randomized positional scoring rules, it still remains an unresolved challenge. For random committee member rules, even the more basic question of identifying the optimal distortion they can achieve remains open, though we are able to pinpoint it to be between $\Omega(m^{2/3})$ and $O(m^{4/5})$. Taking a step back, while distortion is a reasonable theoretical measure for efficiency, it remains to be seen whether it is also correlated for other measures of efficiency one may care about in practice. For example, in the context of food donations, Lee et al. [15], who use the deterministic Borda rule to make decisions, suggest a number of important decision factors other than the social welfare of the stakeholders, such as whether the donations are distributed equitably, how long the drivers have to travel to deliver donations, and whether organizations with higher poverty rates, lower median incomes, and worse food access are receiving sufficient donations. An important next step would be to measure the efficiency of explainable randomized voting rules in real-life applications such as food donation.

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
