# OpenReview forum: "Explainable and Efficient Randomized Voting Rules"
_NeurIPS.cc/2023/Conference — NeurIPS 2023 poster_

### Official Review · Reviewer_c5N6 · 2023-07-04

**Soundness:** 3 good
**Presentation:** 3 good
**Contribution:** 3 good
**Rating:** 6
**Confidence:** 1

**Summary:**

This paper is concerned with introducing a simple randomization "step" to deterministic voting rules so as to increase efficiency, as measured through the distortion framework (and also min-sw). The goal is to retain both the explainability of deterministic voting rules while increasing efficiency.

**Strengths:**

The paper addresses an interesting tension between explainability and efficiency. I commend the author for this well-motivated problem. I think formalizing the tradeoff is definitely an important question.

**Weaknesses:**

Let me preface by saying that voting is not my primary research area, and so, it is not entirely clear to me the extent of the innovation and if the number and quality of the results/techniques are significant for the community.

With that said, some questions come to mind when reading:

1. The fundamental premise of the paper is that deterministic => explainable.
Is this a well-established fact in the voting community (besides appealing to [16] as in L46)? If not, then the whole investigation seems less well-founded.

2. To me, it seems natural to have a knob for introduce the amount of randomization, which can also serve as a measure of how ``simple'' the randomization step is.

For instance, what about a ``soft'' notion of randomization? There is a tunable parameter that introduces a measured amount of randomization. Have the authors considered this and bounding distortion and min-sw as a function of this parameter?



**Questions:**

Please see above.

**Limitations:**

Seems fine.

---

> ### Author Rebuttal · Authors · 2023-08-10
>
> We are sincerely grateful for your time and energy reviewing the paper. To answer your questions:
>
> > The fundamental premise of the paper is that deterministic => explainable. Is this a well-established fact in the voting community (besides appealing to [16] as in L46)? If not, then the whole investigation seems less well-founded.
>
> We would say that in general, yes, most deterministic rules are viewed as more explainable than most randomized rules. Besides [16], we have cited the work of Lee et al. [15], who used deterministic rules in the context of food donations and explained them to the stakeholders. These were deemed understandable by the stakeholders and thus adopted.
>
> Additionally, political elections are almost always deterministic. Indeed, even significantly complicated deterministic rules such as the Schulze method are widely deployed in the real world (see https://en.wikipedia.org/wiki/Schulze_method#Usage). In contrast, even simple randomized rules have no known uses in real-world elections (see https://en.wikipedia.org/wiki/Random_ballot#Prevalence).
>
> At least part of the resistance to using randomized rules comes from a lack of explainability. This is related to the inability to convincingly explain precisely why a decision was taken due to chance being involved; see [C, p107] below. However, when the randomization step is simple, it may be possible to explain why a certain alternative had high score under a randomized positional scoring rule or was chosen to be in the support under a random committee member rule along with the final randomization step.
>
> Therefore, although some deterministic rules can be more explainable than others, we believe that the deterministic-randomized divide in terms of explainability is viewed as more significant.
>
> [C] Duxbury, Neil. Random justice: On lotteries and legal decision-making. Oxford University Press, USA, 2002.
>
>
> > To me, it seems natural to have a knob for introduce the amount of randomization, which can also serve as a measure of how "simple" the randomization step is. For instance, what about a "soft" notion of randomization? There is a tunable parameter that introduces a measured amount of randomization. Have the authors considered this and bounding distortion and min-sw as a function of this parameter?
>
> That sounds like an interesting line of investigation, which we have not pursued. This would require finding an appropriate measure for the "amount of randomization". One could look into entropy or the size of the support; our random committee member rules can be viewed as an example of the latter. Although, such simple measures would likely fail to accurately capture what makes randomization less appealing to humans. Ultimately, we believe that user studies may be needed to come up with the right measures.

---

> > ### Comment · Reviewer_c5N6 · 2023-08-17
> > **Reply**
> >
> > Thanks for the reply, I've updated my score to match the consensus.

---

### Official Review · Reviewer_uYtf · 2023-07-04

**Soundness:** 4 excellent
**Presentation:** 4 excellent
**Contribution:** 3 good
**Rating:** 6
**Confidence:** 4

**Summary:**

The paper studies two types of simple (or in their words *explainable*) randomized voting rules with respect to their (utilitarian) distortion. These are, (1) randomized positional scoring rules where every agent gives a certain number of points to their first ranked, second ranked, etc. alternative and a winning alternative is chosen with probability proportional to the received scores, and (2) random uniform committee member rules, where the rule first selects a committee and then choses a winning alternative uniformly at random from this committee.

The story of the paper is that decision making within AI tools should be both *explainable* and *efficient*. For the sake of explainability, the paper focuses on voting methods and suggests that the above mentioned methods offer a particularly explainable procedure. For measuring efficiency, the paper utilizes the commonly studied distortion framework.

The paper then provides tight bounds on the distortion of members of these families of randomized voting rules. While these bounds show that some of these simple randomized rules do not even reach the same worst-case guarantee as a trivial benchmark (the rule selecting a candidate uniformly at random), the paper presents computational experiments in which the studied rules outperform the benchline (in some cases) as well as deterministic rules.

**Strengths:**

- The paper is certainly well written, and easy to follow. Especially, I appreciated that the papers gives insights into the developed techniques, albeit that most of the proofs had to be deferred to the appendix.
- I see the main strength of the paper in its technical contribution, especially the new technique of deriving distortion bounds via a global lower bound on the social welfare. I am optimistic that these will be helpful for future research on utilitarian distortion.
- I appreciate the breadth of the results, since many of the results hold for entire classes of voting rules.
- The literature on distortion of voting rules has been very active in recent years in social choice, hence I am optimistic that the paper will receive significant attention within the community.

**Weaknesses:**

 I find the story in the introduction more distracting and unfitting than helpful. In particular, the following questions are completely left open: (1) In which AI applications can voting actually serve as a reasonable alternative? (2) In which AI application are we in the situation that a set of agents have cardinal utilities but they can only report *ordinal* preferences? In particular, if the *agents* in the AI application are actually AI systems themselves, the argument that cardinal preferences require too much cognitive effort is not very convincing. (3) If there are AI tools that satisfy the above criteria, which objectives are important in these contexts? (Why is it small distortion?) Personally, I think it would be much more helpful for the reader to shift the focus of the introduction much more towards the content of the paper, e.g., (a) How can we measure explainability in randomized rules and do there exist other rules (besides the two groups) that are explainable? (b) What is the status quo on utilitarian distortion among deterministic and randomized voting rules?
- I am not convinced that the experimental results are very meaningful since the experiments consider only one specific random model (i.e., Mallows Model) which has been shown to be able to produce only a small fraction of all possible types elections (see, e.g., "Putting a Compass on the Map of Elections", Boehmer et al, IJCAI 2021). While this is a general issue in social choice theory, this is particularly relevant for this paper, since the main positive results seem to stem from the fact that the studied voting rules behave well "in practice" (evaluated as their behavior on Mallows) and even within the space of Mallows elections, there is no randomized voting rule which dominates the benchmarks for all Mallows elections. Hence, I think that extending the experimental evaluation to more models would significantly increase the relevance of the experimental part of the paper.

**Minor Comments**
- Related Work section: For the sake of comparison it would have been nice to provide an overview of the best distortion guarantees that can be achieved by deterministic and randomized rules.
- Line 179: sc(a,s) instead of sc(c,s)?
- Line 194: \alpha instead of \epsilon?
- Line 212: "This allows us to we subdivide"
- Line 221: I was missing some context here: Does this hold for every distribution of alternatives?
- Theorem 2 seems unusual. I understand that this was done to save space.
- If I understand correctly, $k$-committee member rules define a family of randomized voting rules. However, Theorem 3 makes it sound as if this would be one specified rule.

**Questions:**

- Have you studied the natural combination of your two (family of) rules? That is, for some scoring vector and some $k$, consider the $k$ alternatives with highest score and then randomize over these with probability proportional to their score. I was wondering whether this rule might be able to inherit the advantages of proportional scoring rules, while reducing the size of the support. Do you know anything about this rule?
- If I understand correctly, in your experiments, you omit the $k$-committee rule from Theorem 4, which was proven to have bounded distortion in the worst case. Why did you decide to do so?
- Why did you decide to focus on Mallow's model in the experiments? Do you have any reason to believe that results for other random models would be similar?


**Limitations:**

While I do think that the authors do a good job in pointing to the limitations of their work within Section 6, I think that the introduction of the paper is in parts misleading and should be tailored more closely to the main part of the paper. Also, the limitations of the experiments could be addressed more transparently.

---

> ### Author Rebuttal · Authors · 2023-08-10
>
> Thank you for your time and helpful comments. Regarding your questions:
>
> > Have you studied the natural combination of your two (family of) rules?
>
> This is a nice idea, which we did not consider. These rules are only slightly less explainable than UR$_k$ rules, so if they lead to significant efficiency gain, it may be worth considering them.
>
> Theoretically, it is not hard to show that most of them also have infinite distortion, just like most UR$_k$ rules, because it is possible for none of the $k$ alternatives in the support to be the top choice of any agent.
>
> Empirically, we ran further experiments to compare these rules (coined PR$_3$) against UR$_3$ rules under Mallows' model; please see Figure 2 in the PDF attached to the common response.
>
> For low $\phi=0.1$, when the rankings are highly correlated, PR$_3$ indeed generally perform better than UR$_3$, which is expected because here the best alternative is often obvious and PR$_3$ can place a higher probability on it. The effect is more pronounced when the best alternative has a much higher score (so, most under plurality, less under Borda/harmonic, almost negligible under 3-approval). However, PR$_3$ Plurality does get worse with large $m$, mimicking D Plurality. Thus, if $\phi$ is low, it may make sense to prefer PR$_3$ Borda/harmonic, and also PR$_3$ Plurality if further $m$ is small.
>
> However, this seems to be the extent of their helpfulness. For the intermediate $\phi=0.5$, all PR$_3$ and UR$_3$ rules have similar performance, and for the impartial culture with $\phi=1$, each PR$_3$ rule coincides with its corresponding UR$_3$ rule as the top 3 alternatives likely have very similar scores (leading PR$_3$ to also pick from them almost uniformly at random).
>
> By the way, once we allow non-uniform randomization, it leads us to the family of randomized rules with support of size $k$; this generalizes both PR$_k$ rules and random committee member rules. Interestingly, our proof of the $m^2/k^2$ lower bound from Theorem 3 actually holds for this broader family, but not the proof of the $k$ lower bound. This leads to the interesting open question whether a randomized rule with support size $k = o(m)$ can achieve near-optimal distortion of $O(\sqrt{m} \operatorname{polylog}(m))$. For random committee member rules, we proved the answer is no (Theorem 3).
>
> We will be happy to add these discussions and experimental results to the appendix of our paper with a reference in the discussion section.
>
> > in your experiments, you omit the $k$-committee rule from Theorem 4...Why did you decide to do so?
>
> That is a good question. We omitted this rule from the experiments for two main reasons.
>
> First, it involves computing an approximately stable committee, which is a computationally difficult task. There is a polynomial time algorithm of Jiang et al. [40], but we suspect it would take long in practice since it relies on using multiplicative weights update to solve a zero-sum game followed by a complicated iterative rounding procedure. One can instead write an integer linear program (ILP), which has bad worst case but perhaps works better in practice.
>
> Second, even if we implemented it, we did not believe that it would perform any better on average than the best of the UR$_3$ rules we already use in our experiments, being a rule designed specifically to get good *worst-case* distortion.
>
> That said, we ran further experiments implementing this rule using the aforementioned ILP to compute approximately stable committees, and compared it in the same Figure 2 of the attached PDF mentioned above. As expected, this rule generally does worse (e.g., markedly so for $\phi=0.5$), although it is interesting that at the extremes of $\phi=0.1$ and $\phi=1$, its performance is similar to the UR$_3$ rules (though never better than the best of the UR$_3$ rules).
>
> > Why did you decide to focus on Mallow's model in the experiments? Do you have any reason to believe that results for other random models would be similar?
>
> We elected to limit the scale of our experiments because we view our extensive theoretical results to be our primary contribution. We chose Mallows' model because it is one of the most popular random ranking models in the computational social choice literature.
>
> That said, we agree that apriori there is no strong reason to believe that results on other random models would be similar. While detailed experiments with more models is an avenue for future work, we ran preliminary experiments with the Polya-Eggenberger Urn Model (from the Map of Elections paper you mentioned) and the Plackett-Luce model, another popular random ranking model. Note that the urn model experiments are limited to $m \le 18$ because sampling from this model is computationally expensive for large $m$ (https://github.com/PrefLib/preflibtools/blob/main/preflibtools/instances/sampling.py).
>
> Figure 1 in the attached PDF shows the results.
>
> Results for the Plackett-Luce model and the urn model with $\alpha=0.01$ are very similar to those for impartial culture from our submission (Mallows' model with $\phi=1$), where randomized positional scoring rules perform better than UR$_3$ rules, which in turn perform better than deterministic rules. Note that the urn model with $\alpha=0$ would be exactly the impartial culture, so this is not very surprising.
>
> Results for the urn model with higher values of $\alpha$ are more interesting. They share some similarities with the Mallows results, such as the order among the three families mentioned above and R Plurality generally performing very well. But there are also crucial differences: for example, in the Mallows experiments, there are no significant differences between the different deterministic rules or between the different UR$_3$ rules, but clear differences show up in the urn experiments, with plurality-based rules performing better.
>
> We will be happy to add these additional plots to the appendix of our paper.

---

> > ### Comment · Reviewer_uYtf · 2023-08-14
> > **Response**
> >
> > I thank the reviewers for their detailed and helpful response, and in particular, for addressing my questions by running more experiments. At this point, I do not have any further questions.

---

### Official Review · Reviewer_VN7K · 2023-07-05

**Soundness:** 4 excellent
**Presentation:** 3 good
**Contribution:** 3 good
**Rating:** 6
**Confidence:** 3

**Summary:**

This paper studies distortions of two family of "explainable" randomized voting rules  -- randomized positional scoring rules and random committee member rules. In the literature of explainable voting rules, various deterministic voting rules have been widely studied and believed to be explainable to some extent. However, recent studies showed that randomized voting rules have efficiency advantages under the measure of distortion. This work aims to combine the best of both worlds -- adding one simple randomization step to explainable deterministic voting rules in order to improve its efficiency. Specifically, randomized positional scoring rules select each alternative with a probability proportional to their scores, and random committee member rules select an alternative uniformly at random among top $k$ candidates with the highest scores.

In this paper, efficiency of certain voting rules is measured by distortion, which assumes each voter has a utility to each agent(normalized to $1$) and reports an ordinal preference that is consistent with her utility. The distortion is then defined as the ratio between expected total utility of the winner and the total utility of the optimal alternative in the worst case.

The paper prove tight distortion bounds for a number of randomized positional scoring rules including Plurality, Borda. Under random committee rules, the authors propose a novel voting rule that achieves sublinear distortion.

Finally, the paper also includes a numerical simulation that demonstrates distortion performance of these two classes of randomized voting rules on synthetic data.

**Strengths:**

- The distortion analyses of randomized positional scoring rules are fairly complete that include asympototically tight bounds.
- The main technical lemma (lemma 2) is novel and mathematically non-trivial that surprisingly tightly captures the lower bounds of the minimal social welfare of a wide range of randomized positional scoring rules.
- It is nice to have some numerical simulations to demonstrate the advantages of randomized voting rules compared to their deterministic counterparts.

**Weaknesses:**

- The introduction tries hard to draw connections between its actual technical contributions (i.e. distortions of randomized voting rules) with literature of explainable AI. However I feel the story a bit disconnected. I understand that certain families of deterministic voting rules have been studied in the context of explainable voting rules. Nevertheless, it is not clear to me why adding one more randomization step won't jeopardize the explainability. For instance, some explainability are based on aximatization, but adding randomization might be incompatible with the original aximatization. I encourage the authors to at least elaborate more and make more convincing arguments on the explainability issue.

- Authors should include more related works about distortions of voting rules and compare their results with other distortion results of different voting rules (deterministic or randomized) in the literature.

- The paper is generally well written, however, to me the main technical lemma (lemma 2) lacks intuition. I think it might be helpful if the authors could expressly explain the quadratic programming for a simple example in the body of the paper.

**Questions:**

- Line 297 mentions "worse than the optimal $\sqrt{m}$". I might be missing something but where does this $O(\sqrt{m})$ bound comes from and what do you mean by "optimal"?

- Could the authors compare randomized propositional voting rules with other randomized voting rules that may be more efficient but less explainable?

---

> ### Author Rebuttal · Authors · 2023-08-10
>
> Thank you for your time and effort in reviewing our paper. Here are our ansewers to your questions:
>
> > Explainable AI...Nevertheless, it is not clear to me why adding one more randomization step won't jeopardize the explainability....I encourage the authors to at least elaborate more and make more convincing arguments on the explainability issue.
>
> You are absolutely correct that when adding randomization, we inevitably lose some explainability. However, our argument, as laid out in the paper, is that in some contexts it may be important to add randomization in order to achieve the significant efficiency gains it provides over deterministic rules. Our point is that instead of adding arbitrary or complex randomization, it may be possible to add simpler and structured randomization, which can retain explainability to some degree while still providing significant efficiency gains. We will clarify this further in our revision.
>
>
> > Line 297 mentions "worse than the optimal ". I might be missing something but where does this bound comes from and what do you mean by "optimal"?
>
> From Ebadian et al. [19], we know that the lowest possible (across all instances with $m$ candidates) distortion achievable by randomized rules is $\Theta(\sqrt{m})$ --- this is what we mean by "optimal" here. In contrast, we know that the best distortion achievable by randomized positional scoring rules is $\Theta(\sqrt{m \log m})$, which is only a sublogarithmic factor worse than $\Theta(\sqrt{m})$. The question we propose in this line, which we resolve negatively later, is whether random committee member rules can also achieve distortion that is only slightly worse than $\Theta(\sqrt{m})$, perhaps even just $O(\sqrt{m} \operatorname{polylog}(m))$.
>
> > Could the authors compare randomized propositional voting rules with other randomized voting rules that may be more efficient but less explainable?
>
>
> Theoretically, the worst-case optimal rule of Ebadian et al. achieves the optimal $\Theta(\sqrt{m})$ distortion, but it uses advanced machinery (e.g., von Neumann's minimax theorem) and is not very explainable. Further, its distortion is only slightly better than the best $\Theta(\sqrt{m \log m})$ distortion that randomized positional scoring rules can achieve (Boutilier et al. [22]). Thus, no other rule (explainable or not) can be significantly better in the worst case. This includes another rule of Boutilier et al. [22] that achieves a distortion of $O(\sqrt{m}\log^{\*} m)$ (where $\log^{\*} m$ is the iterated logarithm), which is even less explainable due to use of advanced LP-duality techniques.
>
> Empirically, we show that some of the explainable rules (e.g., R Plurality) actually achieve distortion that is, on average, very close to the distortion achieved by even the instance-optimal rule. Thus, once again, no rule (explainable or not) can be significantly better on average. When $\phi=1$, all four common randomized positional scoring rules are close to this benchmark of the instance-optimal rule.

---

> > ### Comment · Reviewer_VN7K · 2023-08-17
> > **Update**
> >
> > I thank the authors for the detailed response that addresses all my questions. I have no further questions at this point.

---

### Official Review · Reviewer_Ypnk · 2023-07-06

**Soundness:** 4 excellent
**Presentation:** 4 excellent
**Contribution:** 3 good
**Rating:** 7
**Confidence:** 4

**Summary:**

This paper provides a discussion of and bounds for distortion (distance) and minimum social welfare of randomized additions to several voting rules. The main body of the paper introduces two versions of randomization for positional scoring rules (either using scores as the distribution or selecting a top-k and then using a distribution). The main technical results of the paper are bounds on the distortion of these randomized voting rules and some empirical simulations that point in the direction that single level randomized voting rules may be better in practice.

**Strengths:**

+ The proofs and the methodology of the proofs are very nice and are likely of independent interest to the distortion community.

+ The paper is very well written overall and easy to follow.

+ The results are very nice and the simulations give a nice indication that even though the worst case results are significantly better, that in practice there could be a lot here.

**Weaknesses:**

- The positioning of the paper in terms of "explainable" is a bit weak, to say the least. The limitations address this but the main "explainable" part is basically the tautology that "voting rules are explainable" which, I guess? There are quite a few poly sci and sociology papers that deal with understanding and explainable voting rules, you could have cited at least one of those papers. Likewise there are a number of papers on models for explain-ability that one could tie into.

- Likewise, how is this machine learning? I don't really understand the vague statement "critical decisions" -- there is plenty of literature, new and old, that argues for using voting rules for concrete examples (aggregating classifiers (Cornelio, C., Donini, M., Loreggia, A., Pini, M.S. and Rossi, F., 2021. Voting with random classifiers (VORACE): theoretical and experimental analysis. Autonomous Agents and Multi-Agent Systems, 35(2), p.22.) or making recommendations (Aird, A., Farastu, P., Sun, J., Voida, A., Mattei, N. and Burke, R., 2023. Dynamic fairness-aware recommendation through multi-agent social choice. arXiv preprint arXiv:2303.00968.)) that would maybe ground the paper better in the ML community than the social choice one.

### Minor Issues:

* Footnote one without any citations is really just random and unnecessary, either substantiate it or delete.

* " explain-ability and efficiency simultaneously to some extent." while I'm all for not overstating results this is a bit informal.. maybe "indicate that in some cases..."

* "explainable voting rules in the section in which" --> this just feels like buzzword bingo, you assert all voting rules are explainable so the extra explainable here is not needed.

**Questions:**

? After Theorem 3 there is a discussion of non-worst case results, is there any hope for average case or distributional results that would improve the worst-case bounds for the random k-committee results?

? The empirical results are nice, though I was hoping for a bit more discussion of why, e.g., UR does so well at lower and so bad at higher distortions. There doesn't seem to be a clear winner across all the experiments except maybe R 3-Approval... any more comments here?

? Why don't we just use the optimal voting rule? Is it because you claim it's not explainable? but from the Boutillier framework it's just a weighted rule that one could explain (as easily as any positional scoring rule) -- so I wonder why one doesn't just use the optimal rule framework?

-------
After Rebuttal:

See my comments below, I think this is a good paper and liked the answers given by the reveiwers.

**Limitations:**

Good discussion. Though I wonder if distortion is the right framework here, a more concrete application (e.g., aggregating classifiers or something) and argument as to why distortion is the right measure would strengthen the fit for NeruIPS.

---

> ### Author Rebuttal · Authors · 2023-08-10
>
> Thank you for your review and the time you took to provide such helpful feedback. In response to your questions:
>
> > ? After Theorem 3 there is a discussion of non-worst case results, is there any hope for average case or distributional results that would improve the worst-case bounds for the random k-committee results?
>
> Studying the average-case distortion of rules, e.g., under distributional assumptions, is certainly a very important and interesting problem. While it was briefly studied in the original paper of Boutilier et al. [22], it remains largely unexplored. Even for the vanilla version (without explainability), the state-of-the-art is very recent, from two working papers from 2023, see [A] and [B] below. Future work could certainly build on the models of those papers and study the average-case distortion of random k-committee rules, as the reviewer suggests.
>
> [A] Caragiannis, Ioannis, and Karl Fehrs. "Beyond the worst case: Distortion in impartial culture electorate." arXiv preprint arXiv:2307.07350 (2023).
> https://arxiv.org/pdf/2307.07350.pdf
>
> [B] Gonczarowski, Y. A., Kehne, G., Procaccia, A. D., Schiffer, B., & Zhang, S. (2023). The Distortion of Binomial Voting Defies Expectation. arXiv preprint arXiv:2306.15657.
> http://procaccia.info/wp-content/uploads/2023/05/expdistortion.pdf
>
>
> > ? The empirical results are nice, though I was hoping for a bit more discussion of why, e.g., UR does so well at lower and so bad at higher distortions. There doesn't seem to be a clear winner across all the experiments except maybe R 3-Approval... any more comments here?
>
> We believe you mean "higher values of $\phi$" rather than "higher distortions".
>
> When $\phi$ is small, the rankings produced via Mallows' model are highly correlated. Thus, just the $3$ alternatives that $\text{UR}_3$ randomizes over (for the four scoring vectors) likely cover the top choices of most agents, leading $\text{UR}_3$ to achieve low distortion.
>
> In contrast, when $\phi$ is large, the alternatives are spread out in agents' preference rankings. Thus, it becomes likely that some alternative is ranked slightly lower by many agents, has a low enough score to not be in the top $3$, but has much higher social welfare than the top $3$ alternatives over which $\text{UR}_3$ randomizes.
>
> This is also why deterministic rules perform reasonably well at low $\phi$, but randomization (over many alternatives, not just $3$) becomes important at high $\phi$.
>
> Other than these, R Plurality and R 3-Approval performing reasonably well, and R Plurality outperforming all four deterministic rules across almost all values of $m$ and $\phi$ (see also the figures in the appendix) are the main takeaways of our experiments, as stated in the paper.
>
> We will explain these more thoroughly in our revision.
>
>
> > Why don't we just use the optimal voting rule? Is it because you claim it's not explainable? but from the Boutillier framework it's just a weighted rule that one could explain (as easily as any positional scoring rule) -- so I wonder why one doesn't just use the optimal rule framework?
>
>
> We believe you're referring to the rule of Boutilier et al. [22], which chooses alternatives proportional to harmonic scores with probability $1/2$ and a uniformly random alternative with probability $1/2$, achieving distortion $\Theta(\sqrt{m \cdot \log m})$, only slightly worse than the optimal of $\Theta(\sqrt{m})$. Note that this is not just a weighted rule. As we state in the paper, it can also be viewed as a randomized positional scoring rule, with the score associated with each position $k$ being $\frac{1}{k}+\frac{1}{m}$. As such, it is indeed explainable by our own criterion.
>
> That being said, its scoring vector is not one of the widely recognized ones, such as Borda, approval, or (regular) harmonic. Hence, people may be less willing to adopt it in practice. This is why we focus on providing an in-depth analysis of the distortion of many explainable rules, including the aformentioned recognized ones, so that the designer can choose depending on the application.
>
> Having said that, the worst-case optimal rule within a given family of rules certainly deserves attention. While this is a solved question for randomized positional scoring rules, we propose this as an interesting open question for random committee member rules and provide bounds towards answering it.

---

> > ### Comment · Reviewer_Ypnk · 2023-08-16
> > **Thanks**
> >
> > Thanks for the responses and I'm still happy with paper overall and happy to accept.
> >
> > I think this argument "That being said, its scoring vector is not one of the widely recognized ones, such as Borda, approval, or (regular) harmonic. Hence, people may be less willing to adopt it in practice. This is why we focus on providing an in-depth analysis of the distortion of many explainable rules, including the aformentioned recognized ones, so that the designer can choose depending on the application." is basically hearsay but I'm not going to belabor the point.
> >
> > "people may be less willing to adopt it in practice" is not a good argument -- I would strongly suggest you back off on that claim a bit and just stick with we see how well we can do with the standard scoring rules from the literature. Arguments from authority are much more convincing :-)

---

> > > ### Author Response · Authors · 2023-08-17
> > >
> > > Thank you; when revising the paper, we will make sure that all the arguments we use in the paper come only from authority about standard scoring rules :)

---

> > > > ### Comment · Reviewer_Ypnk · 2023-08-21
> > > > **:-)**
> > > >
> > > > I mean it's that or give me a user study that supports the claim that these scoring rules are more natural or understandable. But I believe that user study can't be completed before the end of the rebuttal period.

---

### Author Rebuttal · Authors · 2023-08-10

We thank all the reviewers for their detailed and helpful reviews. In response to some of the comments, we ran further experiments and included the results in the PDF attached to this common response. We will be happy to incorporate these results as well as the suggestions of the reviewers when we revise the paper.

---

### Decision · Program_Chairs · 2023-09-21

**Decision:**

Accept (poster)

**Comment:**

The reviewers are all positive and acknowledged the neat results and interesting message, while they all raised questions about the high-level approach and arguments on explainability. Eventually, results in paper seems to be interesting in its own right to at least the AGT community, which makes it a good candidate for poster presentation.

We hope the authors can find the reviews helpful. Thanks for submitting to NeurIPS!